# A flat carborane with multiple aromaticity beyond Wade–Mingos' rules

Wei Lu [1], Dinh Cao Huan Do [1] & Rei Kinjo [1✉]

It is widely known that the skeletal structure of clusters reflects the number of skeletal bonding electron pairs involved, which is called the polyhedral skeletal electron pair theory (PSEPT) or Wade and Mingos rules. While recent computational studies propose that the increase of skeletal electrons of polyhedral clusters leads to the flat structure beyond the PSEPT, little experimental evidence has been demonstrated. Herein, we report the synthesis of a $C_2B_4R_4$ carborane **2** featuring a flat ribbon-like structure. The $C_2B_4$ core of **2** bearing 16 skeletal electrons in the singlet-ground state defies both the $[4n + 2]$ Hückel's rule and Baird's rule. Nevertheless, the delocalization of those electrons simultaneously induces two independent π- and two independent σ-aromatic ring currents, rendering quadruple aromaticity.

[1] Division of Chemistry and Biological Chemistry, School of Physical and Mathematical Sciences, Nanyang Technological University, 21 Nanyang Link, Singapore 637371, Singapore. ✉email: rkinjo@ntu.edu.sg

The concept of aromaticity has been of paramount importance in myriad research fields of chemistry since the discovery in 1865 that the benzene molecule is cyclic[1]. Monocyclic conjugated hydrocarbons, namely annulenes, following the $[4n + 2]$ Hückel's rule exhibit aromatic nature, whereas (singlet) annulenes featuring the $4n$ π electrons system are found to be antiaromatic[2–4]. Significantly, this concept is predicted to be reversed in the lowest triplet state on the basis of Baird's rule[5,6]. Thus, π-conjugated molecules with $4n$ π electrons in the triplet state may exhibit aromatic character[7,8]. Recently, the group of Wang and Boldyrev reported that despite the singlet ground state, the $B_{19}^-$ cluster with 12 π electrons exhibits aromaticity[9]. This can be rationalized by the fact that the $B_{19}^-$ cluster possesses two concentric π-aromatic networks, involving a 10 π-electron system and a 2 π-electron system, both of which satisfy the $[4n + 2]$ Hückel's rule, independently. This study indicates that even π-conjugated molecules with $4n$ π electrons in the singlet state may be aromatic if the $4n$ π electrons are split into several $[4n + 2]$ π-electron systems with delocalization. It is believed, however, that such a peculiar chemical bonding situation can only be found in atomic clusters.

The discovery of molecules of type $C_2B_4H_6RR'$ (R, R' = H, Me, $C_3H_7$) by Onak, Williams and Weiss in 1962, is considered to be the beginning of the chemistry of carboranes, a class of boron clusters with the general formula of $[(CH)_a(BH)_mH_b]^c$, mostly having polyhedral structures[10]. It has been recognized over the past several decades that the chemistry of carboranes can be ubiquitously applied to various research fields, ranging from medicinal chemistry, through catalysis, to materials science[11]. Particularly diagnostic of carboranes is the delocalization of the skeletal electrons through multi-center two-electron bonds, rendering the three-dimensional aromatic at the cage moiety[12–15]. The cage framework of carborane clusters can be rationalized by the polyhedral skeletal electron pair theory (PSEPT), also known as Wade–Mingos' rules, which provides the relationship with the skeletal bonding electron pairs (SEPs)[16–18]. The $n$-vertex carboranes with $n$, $(n + 1)$, $(n + 2)$ and $(n + 3)$ SEPs prefer to adopt *hypercloso-*, *closo-*, *nido-* and *arachno-*structures, respectively. Afterward, the *mno* rules, as an extension of the PSEPT, have been reported to predict the structures of condensed polyhedral clusters with a shared triangle face, edge or single vertex[19]. Meanwhile, advanced computational studies propose that the increase of skeletal electrons of polyhedral clusters by reducing the number of substituents leads to considerable electron occupancy in the antibonding skeletal orbitals, which gives rise to flat structures beyond the PSEPT[20]. In this context, the group of Frenking and Hoffmann reported the theoretical studies on a $C_2B_4H_4$ cluster, and concluded that the favored structure is the *hypercloso* octahedron **I** involving two capped carbon atoms (Fig. 1a)[21]. By contrast, Ding and coworkers proposed that the global minima of the $C_2B_4H_4$ cluster is a ribbon-like isomer **II**, which lies 16.0 kcal mol$^{-1}$ lower in energy than **I**[22]. Interestingly, **II**—despite bearing only two π-orbitals, thus a $4n$ π system—is predicted to be aromatic. Notwithstanding the fundamental significance of both PSEPT and aromaticity, however, an experimental proof regarding the nature of the $C_2B_4H_4$ cluster has never to our knowledge been reported, mainly because of the synthetic challenges of small carborane clusters ($n_{vertex} \leq 12$) bearing fewer substituents than their vertices[23].

Herein, we report the synthesis, single-crystal X-ray diffraction, and computational studies of a flat $C_2B_4R_4$ carborane **2**, the latter of which suggest simultaneous electronic delocalization in both π- and σ-orbitals of the $C_2B_4$ scaffold. In addition, a 6-vertex, 7-SEP carborane **3**, which displays an unusual spiral triangle-strip geometry, is also structurally characterized; both compounds thus defy the PSEPT-based structural prediction.

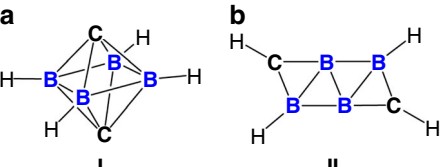

**Fig. 1 Global structures of $C_2B_4H_4$ predicted by theoretical calculations. a** Three-dimensional octahedron **I** with two capped carbon atoms. **b** Two-dimensional ribbon-like structure **II**.

## Results

**Synthesis and structural elucidation of flat carborane 2.** Treatment of tetraatomic boron(0) species **1**[24] with two equivalents of 2,6-diisopropylphenyl isonitrile (DipNC) in benzene at ambient temperature yielded a brown solution, and after workup compound **2** was gained as orange crystals in 23% yield (Fig. 2a). The $^{11}B$ NMR spectrum of **2** exhibits two characteristic broad singlets at 16.5 ppm and 11.3 ppm, which are in line with the computationally estimated $^{11}B$ NMR chemical shifts (the central non-substituted boron: 14.4 ppm, the substituted boron: 9.8 ppm, respectively). The corresponding $^1H$ NMR spectrum shows two downfield singlets at $\delta = 5.7$ ppm ($CH_{BNC3-ring}$) and $\delta = 3.2$ ppm (N$H$), indicative of the hydrogen migration from the $BNC_3$ five-membered ring of **1** to the nitrogen atom of isonitrile during the reaction.

An X-ray diffraction analysis revealed that **2** adopts a $C_2$ symmetry with the central $B_4C_2$ moiety assuming the two-dimensional ribbon-like structure (Fig. 1b). Two $BNC_3$ five-membered ring substituents bounded to the B2 (B2') atoms are in *trans* fashion, which are slightly twisted with respect to the central $C_2B_4$ moiety with the C3–C2–B2–B1' (C3'–C2'–B2'–B1) torsion angle of 18.6(5)°. The $B_4C_2$ core and N1, N1', C2, C2' atoms are nearly coplanar, and the sum of bond angles around B2 (B2') and C1 (C1') are 359.8° and 359.9°, respectively. The C1–B1–B2' angle [177.3°] deviates only slightly from linearity. The B1–B2' (B1'–B2) (1.797(3) Å) and B1–B2 (B1'–B2') (1.709(3) Å) distances are similar to those observed in the small carboranes (1.674–2.089 Å)[25,26]. In contrast, the B1–B1' distance (1.601(4) Å), which falls into the range of typical B–B double bond distances observed in base-stabilized diboranes[27], is slightly shorter than those in the aromatic carboranes (1.625–1.636 Å)[26,28], The B1–C1 (B1'–C1') (1.421(3) Å) and B2–C1 (B2'–C1') (1.501(3) Å) distances are significantly longer than the reported B–C double bond (1.351 Å), but comparable to the typical B–C unsaturated bonds seen in the aromatic carboranes (1.486–1.502 Å)[26,29]. **2** represents a rare example of a molecule involving planar tetracoordinate boron centers[30–32]. Taking these metrics into account, it can be envisaged that the skeletal electrons are delocalized within the $C_2B_4$ core. It is salient to highlight that compound **2** with the 6-vertex and 8-SEP system does not show a *nido* cage structure predicted by the Wade–Mingos' rules but instead adopts a planar geometry[20,22].

**Theoretical studies of 2.** The electronic structure of **2** was further examined by DFT calculations using the optimized structure of the slightly modified model compound **opt-2'**. The singlet ground state is found to be thermodynamically more stable by 26.7 kcal mol$^{-1}$ than the triplet ground state **opt-2'T**, which shows the distorted $C_2B_4$ plane. The two most characteristic molecular orbitals are illustrated in Fig. 3a. The highest occupied molecular orbital (HOMO) of **opt-2'** mainly corresponds to the π-type orbitals over the C1B1B2 (C1'B1'B2') three-membered rings, which exhibits anti-bonding relationship with the C2–C3 (C2'–C3') π-orbitals and the lone pairs of nitrogen atoms. The HOMO−3 comprises the π-type orbitals of the central $B_4$ moiety

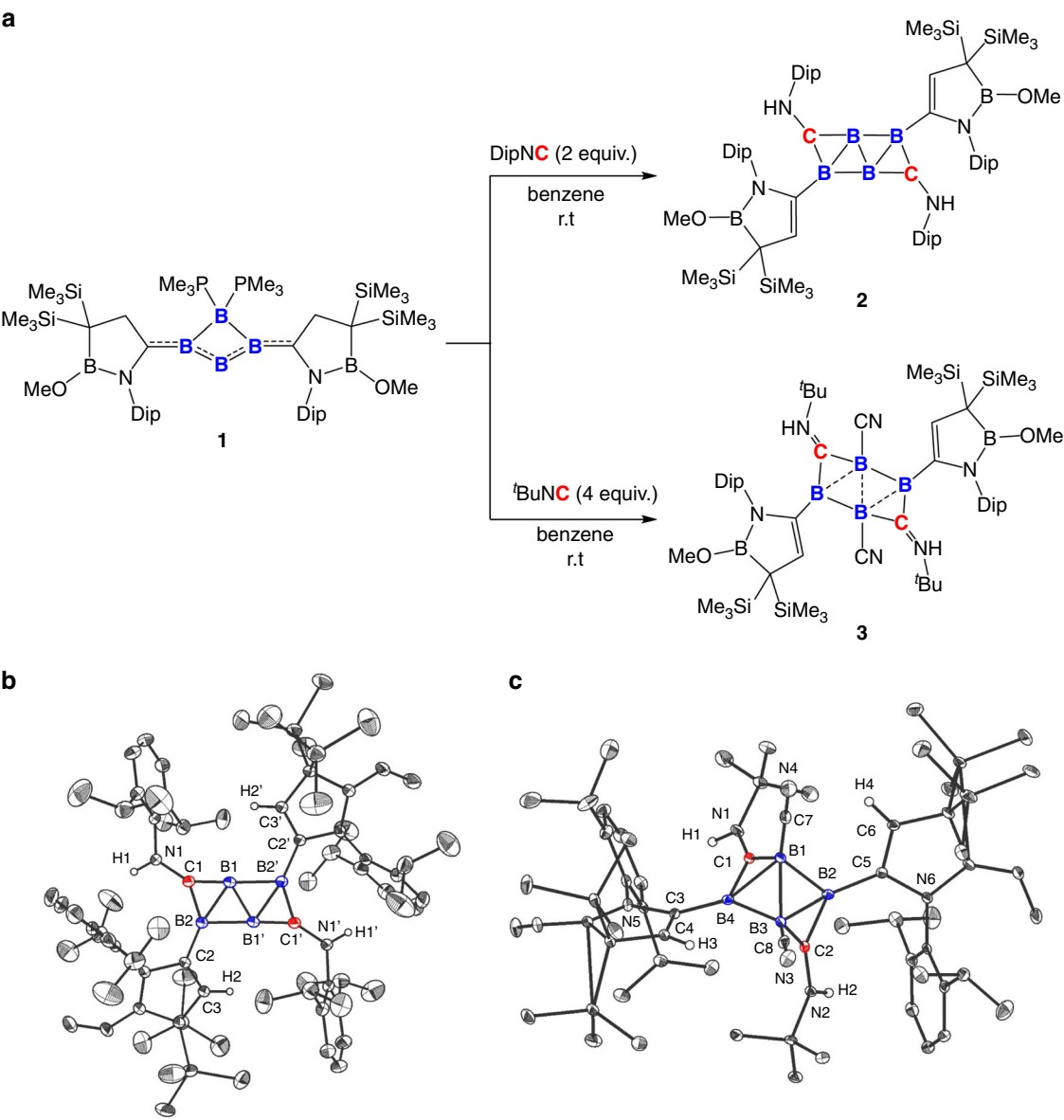

**Fig. 2 Synthesis and characterization of 2 and 3. a** Synthesis of **2** and **3** from tetraatomic boron(0) **1** with isonitriles (Dip = 2,6-diisopropylphenyl, $^t$Bu = tertiary butyl). **b** Solid-state structure of **2**; hydrogen atoms except for those on N1, N1', C3, C3' are omitted for clarity. **c** Solid-state structure of **3**; hydrogen atoms except for those on N1, N2, C4, C6 are omitted for clarity).

with contribution from the 2,6-diisopropyl phenyl rings. The B–C and B–B bonding interactions within the σ-frameworks of the $C_2B_4$ core are well found in the MOs (Supplementary Fig. 13). The 4-electron π-system and 12-electron σ-system indicate that the skeletal electrons are delocalized in both π- and σ-frameworks of the $C_2B_4$ core[22]. The respective Wiberg bond index (WBI) values of 1.3091 for B1–C1 (B1'–C1') and 1.1069 for B2–C1 (B2'–C1') are in line with their bond distances. Interestingly, despite the short B1–B1' distance (1.601(4) Å), a relatively small WBI value of 0.8041 is found for B1–B1', indicating the weak covalent bond character. Thus, the observed short B1–B1' distance is likely due to the strained ribbon-like structure rather than the strong π/σ-bonding interaction.

To assess the delocalization of electrons within the $C_2B_4$ core of **2**, the nucleus-independent chemical shifts (NICS) calculation was performed on **opt-2'** and the relevant compounds **III–V** for comparison (Fig. 3b)[33]. The calculated NICS(1) values of the BBB and CBB rings in **opt-2'** (−8.80 ppm and −11.76 ppm) are less

negative compared with those (**III**: −14.13 ppm, **IV**: −16.14 ppm, **V**: −15.34 ppm) of 2π-aromatic molecules **III–V**, indicating that the π-aromatic nature of **2**, in particular at the BBB rings, is relatively weak (Tables 1 and 2). This can be rationalized by the fact that there are only 4π electrons formally spreading over the $C_2B_4$ plane (Fig. 3a). By contrast, the corresponding NICS(0) values of **opt-2'** (−28.19 ppm and −31.36 ppm) are significantly more negative with respect to those (**III**: −23.24 ppm, **IV**: −15.56 ppm, **V**: −18.38 ppm) estimated for **III–V**, indicating the pronounced σ-aromatic character of **2**. As a proof of concept, we also undertook the analogous NICS analysis for the prototypical molecule **opt-2'(H)**, which showed the more negative NICS(0) values (BBB ring: −20.59 ppm, CBB ring: −30.02 ppm) and less negative NICS(1) values (BBB ring: −9.07 ppm, CBB ring: −15.03 ppm), respectively. This trend is similar to that seen in **opt-2'**, demonstrating a broadly identical electronic relationship between **opt-2'** and **opt-2'(H)**, whereas the observed slight differences in the NICS(0) and NICS(1) values between them

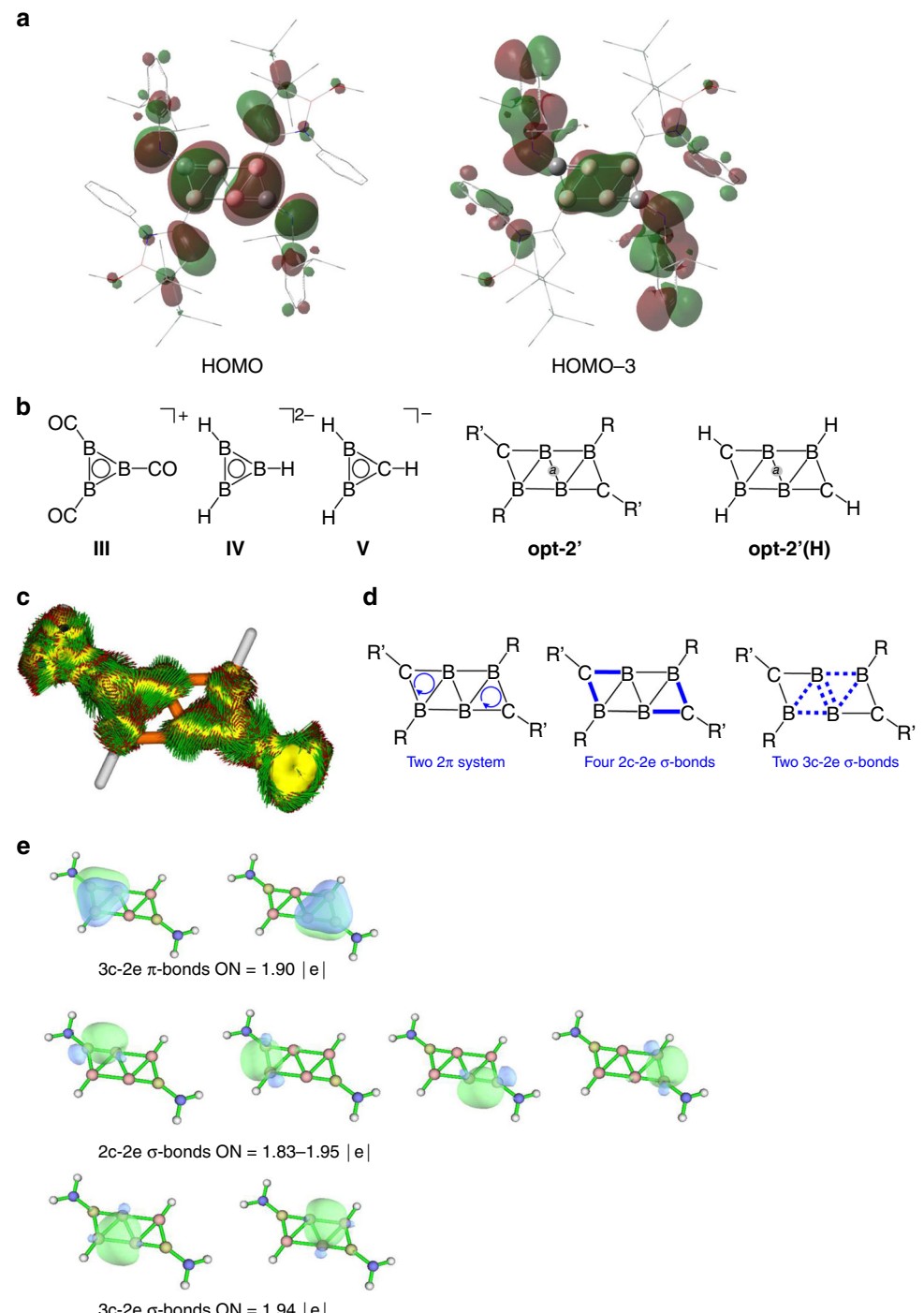

**Fig. 3 Computational studies of 2. a** Plots of the key molecular orbitals of **opt-2'**. **b** Calculated NICS values (in ppm) of **III–V**, **opt-2'**, and **opt-2'(H)** at the B3LYP/6-311G** level of theory. **c** ACID plot of π framework of $C_2B_4H_2(NH_2)_2$ **opt-2''** at an isosurface value of 0.025. **d** Schematic drawings indicating the π- and σ-bonding electrons (blue clockwise arrows and blue solid/dashed lines respectively) in the $C_2B_4$ cores of **2**. **e** Bonding analysis of **opt-2''** using AdNDP method (isosurface value = 0.05).

**Table 1 Calculated NICS values (BBB ring, in ppm) of III–V, opt-2', and opt-2'(H) at the B3LYP/6-311G** level of theory.**

| NICS | III | IV | opt-2' | opt-2'(H) |
|---|---|---|---|---|
| 0 | −23.24 | −15.56 | −28.19 | −20.59 |
| 1 | −14.13 | −16.14 | −8.80 | −9.07 |

**Table 2 Calculated NICS values (CBB ring, in ppm) of V, opt-2', and opt-2'(H) at the B3LYP/6-311G** level of theory.**

| NICS | V | opt-2' | opt-2'(H) |
|---|---|---|---|
| 0 | −18.38 | −31.36 | −30.02 |
| 1 | −15.34 | −11.76 | −15.03 |

**Table 3 Calculated NICS values (at point *a*, in ppm) of opt-2′ and opt-2′(H) at the B3LYP/6-311G** level of theory.**

| NICS | opt-2′ | opt-2′(H) |
|------|--------|-----------|
| 0 | −18.69 | −13.04 |
| 1 | −7.73 | −7.48 |

could be attributed to the substituent effect (Supplementary Fig. 14, Supplementary Table 14)[34]. The NICS(0) and NICS(1) values at the central point (*a*) of the $C_2B_4$ planes in **opt-2′** (−18.69 ppm and −7.73 ppm) and **opt-2′(H)** (−13.04 ppm and −7.48 ppm) support the σ-aromaticity (Table 3). It is noteworthy that both the π-system with four electrons and σ-system with 12 electrons are usually considered as antiaromatic.

The property of electron delocalization was further evaluated by the anisotropy of the current-induced density (ACID) analysis using another structurally simplified model compound $C_2B_4H_2(NH_2)_2$ **opt-2″**[35]. The clockwise current density vectors were plotted on the π-component of ACID isosurface at the two CBB rings (Fig. 3c), which confirms the distinct ring current over the CBB rings, and suggests the presence of the two local aromaticity of the 2π CBB three-membered rings (Fig. 3d, left), each of which satisfies the $[4n + 2]$ electron Hückel's rule independently. This observation agrees with the more negative NICS(1) values of the CBB rings than those of BBB rings (Tables 1 and 2). Indeed, Natural Bond Orbital (NBO) analysis confirms the three-center two-electron (3c-2e) π-bonding over the two CBB rings (Supplementary Tables 15 and 16). We also performed bonding analysis using the adaptive natural density partitioning (AdNDP) method (Fig. 3e), which presents two 3c-2e π-bonding over the two CBB rings[36]. In the σ framework, all C–B bonds are found to be 2c-2e peripheral bonds (Fig. 3d, middle), whereas the delocalized 3c-2e bonding character is seen at the two BBB three-membered rings, each of which—fulfilling the Hückel's rule with $n = 0$—could be responsible for the local σ-aromaticity (Fig. 3d, right). Collectively, the experimental structural analysis and theoretical calculation results suggest that the $C_2B_4$ moiety of **2**, albeit the globally antiaromatic species in general, features two π- and two σ-aromatic distinct frameworks, manifesting the multiple local aromaticity[32,37–41]. It is most likely that the presence of consecutive 3c-2e π and 3c-2e σ bonding plays a pivotal role in the stabilization of the flat ribbon-like geometry[42–45].

**Synthesis and structural analysis of spiral carborane 3.** Interestingly, when compound **1** was reacted with four equivalents of *tert*-Butyl isonitrile (*t*BuNC) in benzene at ambient temperature, a deltahedron carborane derivative **3** was obtained in 26% isolated yield (Fig. 2a). The $^{11}B$ NMR spectrum of **3** shows broad singlets at $\delta = -12.2$ ppm for B–C≡N and at $\delta = 12.2$ ppm for BBNC3-ring, respectively. Orange single crystals of **3** were obtained by recrystallization from a saturated toluene solution, and the solid-state structure determined by X-ray diffraction analysis shows that **3** with the 7-SEP system features a spiral triangle-strip structure (Fig. 2c), rather than octahedron predicted for the 6-vertex and 7-SEP *closo*-carboranes according to Wade–Mingos' rules. The solid-state structure of **3** also differs from trigonal bipyramidal and butterfly shapes corresponding to the global minima of the substituted dicarboranes $(C_2B_4R_6)$[46], suggesting the pronounced kinetic effect by the bulky substituents of **3**. The central $C_2B_4$ moiety is significantly twisted with the C2–B2–B3–B1, B2–B1–B3–B4 and B3–B1–B4–C1 torsion angles of 144.5°, 117.4°, and 141.1°, respectively. The B–B

distances of B1–B2 (1.767(3) Å), B1–B3 (1.765(3) Å), B1–B4 (1.852(3) Å) B2–B3 (1.863(3) Å), and B3–B4 (1.762 (3) Å) fall into the typical range of B–B bonds (1.674–2.089 Å) reported for the small carborane clusters. The B–C distances of B1–C1 (1.545 (3) Å), B2–C2 (1.529(3) Å), B3–C2 (1.540(3) Å), and B4–C1 (1.517(2) Å) are comparable to those found in small carboranes (1.512–1.817 Å)[25,26]. The observed structural change from **2** to **3** confirms that the removal of two electrons from the $C_2B_4$ skeleton may decrease the electron delocalization and therefore the stability of the flat core, which leads to distortion from the planar ribbon-like geometry, consistent with Jemmis' prediction[20]. We have also investigated the reaction of **1** and CO, which however gave a complex mixture and no identifiable product was obtained from the mixture.

## Discussion

Almost five decades after the establishment of the polyhedral skeletal electron pair theory, this work shows that a flat carborane featuring a ribbon-like architecture **2** based on the 6-vertex and 8-SEP system can be isolated. Theoretical studies manifest the delocalization of the skeletal electrons in both π- and σ-orbitals of the $C_2B_4$ core in **2**, inducing the simultaneous π- and σ-ring currents. The synthesis of a relevant carborane **3** with 6 vertices and 7 SEPs is also feasible, which is found to exhibit a spiral triangle-strip geometry. Both compounds **2** and **3** represent the first $C_2B_4$ carboranes disobeying the structural prediction by PSEPT, which may contribute to extending Wade–Mingos' rules and rationalizing the electronic structure of other $E_nR_x$ ($n \geq x$) carborane isomers.

## Methods

**Materials.** For details of spectroscopic analyses of compounds in this paper, see Supplementary Figs. 1–12. For details of density functional theory calculations, see Supplementary Figs. 13–14, Supplementary Tables 2–16 and Supplementary Methods. For details of X-ray analysis, see Supplementary Table 1, and Supplementary Methods.

**General synthetic procedure.** All reactions were performed under an atmosphere of dry nitrogen or argon by using standard Schlenk or dry box techniques; solvents were dried over Na metal, K metal or $CaH_2$. $^1H$, $^{11}B$ and $^{13}C$ NMR spectra were obtained with Bruker AVIII 400 MHz BBFO and JEOL ECA400 spectrometer at 298 K unless otherwise stated. NMR multiplicities are abbreviated as follows: s = singlet, d = doublet, t = triplet, sep = septet, m = multiplet, br = broad signal. Coupling constants *J* are given in Hz. HRMS spectra were obtained at the Mass Spectrometry Laboratory at the Division of Chemistry and Biological Chemistry, Nanyang Technological University. Melting point was measured with an OpticMelt Stanford Research System. Fourier transform infrared (FT-IR) spectra were recorded on a Bruker ALPHA-Transmittance FT-IR Spectrometer. Cyclic voltammetry (CV) was performed on a Biologic SP-50 electrochemical analyzer in anhydrous THF containing recrystallized tetra-n-butyl-ammoniumhexafluorophosphate (TBAPF$_6$, 0.1 M) as supporting electrolyte at 298 K under an inert atmosphere of argon. A conventional three electrode cell was used with a glassy carbon electrode as the auxiliary electrode, silver/silver nitrate as the reference electrode, and a platinum wire as the working electrode. Compound **1** was synthesized according to the literature method[24].

**Synthesis of compound 2.** At ambient temperature, benzene-$d_6$ (2 ml) was added to **1** (0.20 g, 0.20 mmol) to give a red solution, followed by 2 equivalents of 2,6-diisopropyl isonitrile (0.075 g, 0.40 mmol). The reaction mixture was stirred at room temperature for 24 h to give a brown solution. After completion of the reaction, all volatiles were removed under reduced pressure to afford a brown mixture. Slow evaporation of a pentane solution of this mixture in glove box yielded **2** as orange crystals (0.056 g, 23%). **M.p.**: 285 °C (Dec.); **$^1H$ NMR** (400 MHz, $C_6D_6$, 25 °C) δ 7.13–7.10 (m, 4H, Ar*H*), 7.05–6.93 (m, 8H, Ar*H*), 5.69 (s, 2H, CH$_{BNC3-ring}$), 3.36 (sep, *J* = 6.8 Hz, 4H, C*H*), 3.23 (b, 2H, N*H*), 3.20 (sep, *J* = 6.8 Hz, 4H, C*H*), 3.01 (s, 6H, OC*H$_3$*), 1.31 (d, *J* = 6.8 Hz, 12H, CH(C*H$_3$*)$_2$), 1.28 (d, *J* = 6.9 Hz, 12H, CH(C*H$_3$*)$_2$), 1.23 (d, *J* = 6.9 Hz, 12H, CH(C*H$_3$*)$_2$), 1.16 (d, *J* = 6.9 Hz, 12H, CH(C*H$_3$*)$_2$), 0.16 (s, 36H, Si(C*H$_3$*)$_3$); **$^{11}B$ NMR** (128 MHz, $C_6D_6$, 25 °C) δ 35.60 (br), 16.48 (br), 11.25 (br); **$^{13}C$ NMR** (101 MHz, $C_6D_6$, 25 °C) δ 147.2 (*C*-Ar), 145.9 (*C*-Ar), 141.3 (*C*-Ar), 140.3 (*C*-Ar), 132.7 (CH$_{BNC3-ring}$), 127.5 (CH-Ar), 127.0 (CH-Ar), 124.2 (CH-Ar), 123.9 (CH-Ar), 52.2 (OCH$_3$), 28.0 (CH), 27.3 (CH), 25.4 (CH(CH$_3$)$_2$), 24.2 (CH(CH$_3$)$_2$), 23.8 (CH(CH$_3$)$_2$), 23.7 (CH(CH$_3$)$_2$), 0.9 (Si

$(CH_3)_3$)), not observed boron-bounded (NCB, $(TMS)_2CB$ and DipNHC); **FT-IR** (solid, $cm^{-1}$): 3384, 2968, 2873, 1629, 1586, 1521, 1469, 1452, 1374, 1336, 1301, 1244, 1193, 1136, 1101, 1050, 1028, 1007, 929, 864, 838, 821, 782, 756, 708, 678, 622; **HRMS** (ESI): m/z calcd for $C_{70}H_{115}B_6N_4O_2Si_4$: 1221.8656 $[(M+H)]^+$; found: 1221.8654.

**Synthesis of compound 3**. At ambient temperature, benzene-$d_6$ (2 ml) was added to **1** (0.10 g, 0.10 mmol) to give a red mixture, followed by 4 equivalents of *tert*-butyl isonitrile (0.033 g, 0.40 mmol). The reaction mixture was stirred at room temperature for 24 h to give a brown solution. After completion of the reaction, all volatiles were removed under reduced pressure to afford a brown mixture. Slow evaporation of a toluene solution of the mixture in glove box afforded **3** as orange crystals (0.032 g, 26%). **M.p.**: 223 °C (Dec.); **$^1$H NMR** (400 MHz, $C_6D_6$, 25 °C) δ 7.39 (s, 2H, $CH_{BNC3-ring}$), 7.03–6.90 (m, 6H, Ar*H*), 5.21 (s, 2H, N*H*), 3.26 (sep, $J = 6.8$ Hz, 4H, C*H*), 3.04 (s, 6H, OC*H$_3$*), 1.26 (d, $J = 6.8$ Hz, 6H, CH(C*H$_3$*)$_2$), 1.22 (d, $J = 6.9$ Hz, 6H, CH(C*H$_3$*)$_2$), 1.07 (s, 18H, C(C*H$_3$*)$_3$), 1.06 (d, $J = 7.7$ Hz, 6H, CH (C*H$_3$*)$_2$), 1.00 (d, $J = 6.9$ Hz, 6H, CH(C*H$_3$*)$_2$), 0.34 (s, 18H, Si(C*H$_3$*)$_3$), 0.30 (s, 18H, Si(C*H$_3$*)$_3$); **$^{11}$B NMR** (128 MHz, THF-$d_8$, 25 °C) δ 36.84 (br), 12.17 (br), −12.21 (br); **$^{13}$C NMR** (101 MHz, THF-$d_8$, 25 °C) δ 149.2 (*C*-Ar), 147.3 (*C*-Ar), 142.6 (*C*-Ar), 139.3 ($CH_{BNC3-ring}$), 129.6 (*C*H-Ar), 126.4 (*C*H-Ar), 125.1 (*C*H-Ar), 58.2 (*C*(CH$_3$)$_3$), 54.3 (O*C*H$_3$), 31.3 (C(*C*H$_3$)$_3$), 29.8 (*C*H), 28.9 (*C*H), 27.4 (*C*H(CH$_3$)$_2$), 26.2 (CH(*C*H$_3$)$_2$), 25.0 (CH(*C*H$_3$)$_2$), 23.6 (CH(*C*H$_3$)$_2$), 1.8 (Si(*C*H$_3$)$_3$), 1.7 (Si (*C*H$_3$)$_3$), not observed boron-bounded (NCB, $(TMS)_2CB$, $^tBuNHC$ and BCN); **FT-IR** (solid, $cm^{-1}$): 3344, 3301, 2969, 2873, 1562, 1518, 1467, 1450, 1370, 1339, 1298, 1247, 1195, 1134, 1078, 1006, 927, 835, 799, 759, 683, 623; **HRMS** (ESI): m/z calcd for $C_{56}H_{99}B_6N_6O_2Si_4$: 1065.7465 $[(M+H)]^+$; found: 1065.7476.

## Data availability

Metrical data for the solid-state structures of **2** and **3** in this paper have been deposited at the Cambridge Crystallographic Data Centre under reference numbers CCDC 1986134–1986135, respectively. Copies of the data can be obtained free of charges from www.ccdc.cam.ac.uk/structures/. All other data supporting the findings of this study are available within the article and its Supplementary Information.

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

## Acknowledgements
We gratefully acknowledge financial support from Nanyang Technological University and The Singapore Ministry of Education (MOE2018-T2-2-048(S)).

## Author contributions
Experiments and theoretical studies were conducted by W.L. and D.C.H.D. R.K. conceived the idea, and the project was directed by R.K. The paper was written by W.L., D.C.H.D., and R.K.

## Competing interests
The authors declare no competing interests.
