## [Peer Review File · Nature Communications]

REVIEWER COMMENTS

Reviewer #1 (Remarks to the Author):

1. What are the major claims of the paper?

The authors synthesized a novel planar carborane molecule with 16 valence electrons, i.e., C₂B₄R₂R'₂. A thorough bonding analysis revealed that the core of the compound contains four-fold aromaticity that accounts for its unique stability. Thus, it presents the first 4npi example that contrasts the known theories, i.e., Wade-Mingos rule and Baird's rule.

2. Are they novel and will they be of interest to others in the community and the wider field?

The global nature (computationally predicted in ref 22) and the present synthetic availability of 2 lay the foundation for such multiply aromatic species in carboranes chemistry. It would significantly advance future search for more examples that are of the non-Wade-Mingos type. The work would have impact on the carboranes chemistry, and Wade-Mingos rule, Baird's rule and 4n+2 Huckel rule.

3. If the conclusions are not original, it would be helpful if you could provide relevant references.

4. Is the work convincing, and if not, what further evidence would be required to strengthen the conclusions?

Yes.

5. On a more subjective note, do you feel that the paper will influence thinking in the field?

Yes. The work is quite impressive since up to date nearly all experimentally reported carboranes are of the higher member usually with more than 10 vertexes. The unpleasing situation could surely be ascribed to the great difficulty in synthesis (e.g., finding suitable synthetic precursor and routes). The present laboratory synthesis of such a "small" core (only 6 vertexes) with exotic characteristics marks the opening of studying such interesting carboranes. The present work clearly showed that besides the well-known Wade-Mingos topology, planar carboranes with 4n electrons in multiple aromaticity ($\sigma + \pi$) can be sufficiently stable and synthesized.

In view of the above, I strongly recommend its acceptance in Nature Communications if the following revisions are made.

1. On page 9, the comparisons of NICS(0) and NICS(1) values are made between the synthesized molecule and several small model molecules. Since IV and V both contain the hydrogen substituents, I suggest the authors to include the discussion of the target planar molecule also with substituent H. In this way, it would be more balanced to make comparisons between all H-substituted species. Besides, one can get information about how the substituent R and R' influence the NICS values of 2. I hope to see the discussions how the bulky substituents influence the structural and electronic features of the 2. Since in actual synthesis, the non-hydrogen substituents have to be utilized for protection, such knowledge would be very useful.

2. Besides compound 2, the authors also synthesized a compound 3 that belong to the family the C₂B₄R₆. A very recent work (Chem. Commun. 2019, 55, 6373-6376) has shown that suitably peri-substituted C₂B₄R₆ can have completely different global structures (i.e., trigonal or butterfly) from the closo-form predicted by the Wade-Mingos' rules. The structure of compound 3 is surely different from those predicted by Wade-Mingos rules and the ref (Chem. Commun. 2019, 55, 6373-6376). I feel that compound 3 may not be the global structure, and its synthetic availability could ascribed to the kinetic hindrance from isomerization to other isomers. The reference should be cited and compared. Anyway, compound 3 is a nice example to demonstrate the structural diversity in carboranes chemistry.

Reviewer #2 (Remarks to the Author):

Authors report for the first time the synthesis of a planar 6 vertex dicarborane violating the Wade-Mingos rules and therefore I recommend its publication.

Reviewer #3 (Remarks to the Author):

This manuscript describes a very exciting and provocative discovery in the area of carborane chemistry, namely the first example of a planar carborane synthesized by very unusual and novel chemistry. The work is very complete with detailed characterization of the two interesting new compounds by X-ray crystallography and nuclear magnetic resonance. Relevant density functional theory studies are also provided as well as demonstration of aromaticity by nucleus independent chemical shift methods.

In reading this manuscript I wondered what type of product might be obtained by reaction of the original tetraborane with carbon monoxide in view of the close relationship between isocyanides and carbon monoxide.

Publication of this seminal paper in Nature Communications is recommended after considering the following minor points:

- Energies should be rounded from two decimal places to a single decimal place reflecting the accuracy of calculations of this type.
- p. 4: change "experimental prove" to "experimental proof."

Point-by-point response to the reviewers' comments

First of all, we sincerely express our gratitude to all three reviewers for their time, valuable comments, and suggestion.

Reviewer #1:

1. What are the major claims of the paper?

The authors synthesized a novel planar carborane molecule with 16 valence electrons, i.e., C₂B₄R₂'₂. A thorough bonding analysis revealed that the core of the compound contains four-fold aromaticity that accounts for its unique stability. Thus, it presents the first 4n_{pi} example that contrasts the known theories, i.e., Wade-Mingos rule and Baird's rule.

2. Are they novel and will they be of interest to others in the community and the wider field?

The global nature (computationally predicted in ref 22) and the present synthetic availability of 2 lay the foundation for such multiply aromatic species in carboranes chemistry. It would significantly advance future search for more examples that are of the non-Wade-Mingos type. The work would have impact on the carboranes chemistry, and Wade-Mingos rule, Baird's rule and 4n+2 Huckel rule.

3. If the conclusions are not original, it would be helpful if you could provide relevant references.

4. Is the work convincing, and if not, what further evidence would be required to strengthen the conclusions?

Yes.

5. On a more subjective note, do you feel that the paper will influence thinking in the field?

Yes. The work is quite impressive since up to date nearly all experimentally reported carboranes are of the higher member usually with more than 10 vertexes. The unpleasing situation could surely be ascribed to the great difficulty in synthesis (e.g., finding suitable synthetic precursor and routes). The present laboratory synthesis of such a "small" core (only 6 vertexes) with exotic characteristics marks the opening of studying such interesting carboranes. The present work clearly showed that besides the well-known Wade-Mingos topology, planar carboranes with 4n electrons in multiple aromaticity (sigma+pi) can be sufficiently stable and synthesized.

In view of the above, I strongly recommend its acceptance in Nature Communications if the following revisions are made.

(1) On page 9, the comparisons of NICS(0) and NICS(1) values are made between the synthesized molecule and several small model molecules. Since **IV** and **V** both contain the hydrogen substituents, I suggest the authors to include the discussion of the target planar molecule also with substituent H. In this way, it would be more balanced to make comparisons between all H-substituted species. Besides, one can get information about how the substituent R and R' influence the NICS values of **2**. I hope to see the discussions how the bulky substituents influence the structural and electronic features of the **2**. Since in actual synthesis, the non-hydrogen substituents have to be utilized for protection, such knowledge would be very useful.

We have included the hydrogen-substituted compound **opt-2'(H)** in Figure 3(b) for comparison.

We have added a brief discussion stating the influence of the substituents on the NICS values in the main text on page 8.

The further details indicating the structural and electronic features of the **opt-2** and **opt-2'(H)** have also been added in the Supporting Information (Supplementary Figs 14, Supplementary Tables 3 and 14, Supplementary Reference. 51).

Moreover, a relevant paper indicating the influence of the substituents on the electronic features of aromatic molecules has been added as ref 34. Accordingly, the reference number has been updated thereafter.

(2) Besides compound **2**, the authors also synthesized compound **3** that belong to the family of $C_2B_4R_6$. A very recent work (*Chem. Commun.* 2019, 55, 6373-6376) has shown that suitably peri-substituted $C_2B_4R_6$ can have completely different global structures (i.e., trigonal or butterfly) from the closo-form predicted by the Wade-Mingos' rules. The structure of compound **3** is surely different from those predicted by Wade-Mingos rules and the ref (*Chem. Commun.* 2019, 55, 6373-6376). I feel that compound **3** may not be the global structure, and its synthetic availability could ascribed to the kinetic hindrance from isomerization to other isomers. The reference should be cited and compared. Anyway, compound **3** is a nice example to demonstrate the structural diversity in caboranes chemistry.

We thank the reviewer for highlighting that the structure of compound **3** differs from the global minima of the $C_2B_4R_6$ molecule shown in the reference (*Chem. Commun.* **2019**, 55, 6373-6376).

The reference (*Chem. Commun.* **2019**, 55, 6373-6376) has been added as ref. 46, and we have placed the following sentence in the main text on page 11:

“The solid-state structure of **3** also differs from trigonal bipyramidal and butterfly shapes corresponding to the global minima of the substituted dicarboranes (C₂B₄R₆),⁴⁶ suggesting the pronounced kinetic effect by the bulky substituents of **3**.”

Reviewer #2:

Authors report for the first time the synthesis of a planar 6 vertex dicarborane violating the Wade-Mingos rules and therefore I recommend its publication.

We thank the reviewer for highlighting the remarkable point of the result.

Reviewer #3:

This manuscript describes a very exciting and provocative discovery in the area of carborane chemistry, namely the first example of a planar carborane synthesized by very unusual and novel chemistry. The work is very complete with detailed characterization of the two interesting new compounds by X-ray crystallography and nuclear magnetic resonance. Relevant density functional theory studies are also provided as well as demonstration of aromaticity by nucleus independent chemical shift methods.

(1) In reading this manuscript I wondered what type of product might be obtained by reaction of the original tetraborane with carbon monoxide in view of the close relationship between isocyanides and carbon monoxide.

We have investigated the reaction of tetraborane **1** and CO as well, which however gave a complex mixture and no identifiable product was obtained from the mixture.

We added a sentence describing this information in the main text on page 12.

Publication of this seminal paper in Nature Communications is recommended after considering the following minor points:

(2) Energies should be rounded from two decimal places to a single decimal place reflecting the accuracy of calculations of this type.

We have revised the decimal accordingly.

(3) p. 4: change “experimental prove” to “experimental proof.”

We have revised the word accordingly.

REVIEWERS' COMMENTS:

Reviewer #1 (Remarks to the Author):

The presently revised manuscript in form of the combined computational and experimental studies on a novel dicarborane compound has sufficiently replied to my concerns. I'm very happy to recommend its acceptance as it is.